# Can Different Fermentation Boxes Improve the Nutritional Composition and the Antioxidant Activity of Fermented and Dried Floodplain Cocoa Beans in the Brazilian Amazon?

**DOI:** 10.3390/foods14081391

**Published:** 2025-04-17

**Authors:** Sabrina Oriana de Souza Begot da Rocha, Maria do Perpétuo Socorro Progene Vilhena, Jesus Nazareno Silva de Souza, César R. Balcázar-Zumaeta, Efraín M. Castro-Alayo, Alexa J. Pajuelo-Muñoz, Braian Saimon Frota da Silva, Maria José de Souza Trindade, Gilson C. A. Chagas-Junior, Nelson Rosa Ferreira

**Affiliations:** 1Laboratory of Biotechnological Processes (LAPROBIO), Graduate Program in Food Science and Technology (PPGCTA), Institute of Technology (ITEC), Federal University of Pará (UFPA), Belém 66075-110, Pará, Brazil; sabrinabegot88@gmail.com (S.O.d.S.B.d.R.); jsouza@ufpa.br (J.N.S.d.S.); braiansaimon@yahoo.com.br (B.S.F.d.S.); chagasjunior.gca@gmail.com (G.C.A.C.-J.); 2Food Science and Technology (CTA), Federal Rural University of Amazonia (UFRA), Belém 66077-830, Pará, Brazil; sprogene@hotmail.com; 3Center for Valorization of Amazonian Bioactive Compounds (CVACBA), Federal University of Pará (UFPA), Belém 66075-110, Pará, Brazil; 4Instituto de Investigación, Innovación y Desarrollo para el Sector Agrario y Agroindustrial (IIDAA), Facultad de Ingeniería y Ciencias Agrarias, Universidad Nacional Toribio Rodríguez de Mendoza de Amazonas, Chachapoyas 01001, Peru; cesar.balcazar@untrm.edu.pe (C.R.B.-Z.); efrain.castro@untrm.edu.pe (E.M.C.-A.); 7255974661@untrm.edu.pe (A.J.P.-M.); 5Socio-Environmental and Water Resources Institute (ISARH), Federal Rural University of Amazon (UFRA), Belém 66077-830, Pará, Brazil; maria.trindade@ufra.edu.br

**Keywords:** *Theobroma cacao*, cocoa fermentation, sensory attributes, biochemical transformations, nutritional composition

## Abstract

This study evaluated the impact of different fermentation boxes on the nutritional and antioxidant composition of dried lowland cocoa beans (*Theobroma cacao* L.), a characteristic product of some producers in the Amazon region. The analysis included ash content, moisture content, pH, titratable acidity, proteins, lipids, flavonoids, antioxidant activities (DPPH, ABTS, and FRAP), and mineral composition. Four types of fermentation boxes were assessed: a projected hexagonal box (PHB), square box (SB), basket (HP), and local square box (LSB). Statistical analyses included ANOVA, Tukey’s test, and Fisher’s LSD test to compare mean differences, while Principal Component Analysis (PCA) identified key contributors, including potassium and magnesium. Spearman correlation analysis revealed significant relationships between soil and almond nutrient profiles. The HP bed exhibited superior phenolic concentration, antioxidant activity, centesimal composition, and potassium and magnesium content. Despite its shorter fermentation period, the LSB bed met quality standards, while the PHB and SB showed intermediate results. Mineral analysis confirmed no toxicological risks, suggesting the beans are safe and enriched with floodplain minerals. These findings emphasize the importance of fermentation methods in determining cocoa bean quality and provide a framework for optimizing processes to enhance their nutritional and functional properties.

## 1. Introduction

The diversity of the cocoa (*Theobroma cacao*) trade is highlighted by its versatility within the bioeconomic landscape [1,2], enabling the application of advanced technologies for extensive exploration and strategic redevelopment in the cocoa cultivation sector. These advancements encompass a wide range of opportunities, from genetic improvements in cocoa varieties to the improvement of cocoa production in producing countries [3].

The state of Pará leads Brazil’s cocoa production, contributing 49% of the national output, followed by Bahia at 45%. Cocoa production in Pará has exhibited sustained growth, supported by investments in sustainable practices, such as agroforestry systems, which simultaneously enhance productivity and preserve the Amazon rainforest [4]. This expansion has primarily occurred in previously degraded areas, contributing to land restoration and reducing deforestation. By 2030, Brazil aims to further increase its cocoa production to over 440,000 tons annually, positioning itself among the world’s top producers, potentially surpassing nations such as Nigeria and Cameroon [5].

Native to the Amazon region, cocoa thrives in the floodplains of the *Baixo Tocantins* region in Pará and is a vital source of agricultural production [6]. Approximately 1600 tons of fermented and dried cocoa beans are produced annually, supporting over 6000 small producers in the cities of Pará like Cametá, Mocajuba, Baião, Limoeiro do Ajuru, and Igarapé-Mirim. This production not only sustains the local economy but also underscores the region’s integral role in the global cocoa trade [7].

Fermentation is an essential stage in cocoa processing, shaping the chemical composition and sensory attributes of cocoa beans. The microbial activity of some yeasts and bacterial species during this process converts mucilage sugars into organic acids, alcohols, and aromatic compounds, influencing flavor and aroma development [8]. Fermentation also affects nutrient retention and the degradation of phenolic compounds, which are essential for antioxidant capacity and health benefits. Recent studies emphasize the role of specific fermentation methods in modulating phenolic profiles and enhancing the nutritional quality of cocoa-derived products [9]. Metagenomic analyses have further provided insights into the microbial dynamics driving these transformations, highlighting the need to optimize fermentation practices to balance flavor development with the preservation of bioactive compounds [10,11,12,13].

In Brazil, the Executive Committee of the Cocoa Crop Plan (CEPLAC) recommends that cocoa fermentation be carried out in wooden boxes, known as *fermentation boxes,* for a minimum of five days and a maximum of seven days. This timeframe is suggested to ensure the proper formation of flavor and aroma precursor compounds while preventing seed death. If the fermentation in these boxes is too short, the development of these sensory characteristics may be compromised, whereas excessive fermentation can lead to beans with a deep brown color and a strong ammonia-like aroma [14,15].

According to CEPLAC recommendations, the fermentation boxes are designed with removable dividers to facilitate ranges from 6 to 10 mm in diameter, spaced approximately 15 cm apart, which allow for the drainage of the liquid generated during the fermentation process [15].

Drying complements fermentation as another critical stage in cocoa processing, regulating fungal proliferation, enhancing structural resilience, and driving biochemical changes that refine flavor, aroma, and color. These transformations are key to achieving the desired quality in cocoa-derived products [10,11,12,16,17].

Studies have shown that cocoa possesses superior antioxidant capacity, surpassing products such as tea and red wine in phenolic compound content [10,18]. However, the concentration of these compounds is inversely correlated with reductions in astringency and bitterness in cocoa beans [18]. This natural decline during fermentation and drying enhances the sensory attributes of cocoa, positioning it as a highly valued and widely sought-after ingredient in the food industry [1,5].

One of the key factors driving consumer acceptance of cocoa products is their sensory attributes, which are closely tied to flavor, aroma, and texture [19,20]. With its unique sensory and nutritional qualities and high energy value, cocoa positively impacts human metabolism, establishing itself as a cornerstone ingredient in the food industry [21,22,23]. Globally sought after, cocoa from the Amazon region of Pará stands out for its exceptional flavor, genetic diversity, origin, and traditional processing methods [21].

The expansion of cocoa production in northern Brazil, particularly in the floodplain islands, has spurred research to improve the quality and marketability of cocoa beans. These efforts aim to enhance value for local producers and to meet the growing demand regionally and globally. This study aimed to investigate whether fermentation methods influence the nutrient and antioxidant composition of cocoa beans throughout the fermentation and drying processes. Specifically, it is hypothesized that different fermentation technologies significantly impact the nutritional and phenolic composition of cocoa beans, with the choice of the fermentation bed playing a critical role in the retention of nutrients and bioactive compounds.

## 2. Material and Methods

### 2.1. Chemicals and Reagents

All the solvents, including ethanol and acetonitrile HPLC grade, were purchased from Sigma-Aldrich (St. Louis, MO, USA). Ultrapure water was obtained from a Milli-Q system (Millipore, Bedford, MA, USA). Acetic acid (≥99%) was sourced from Dinâmica (São Paulo, Brazil).

Standards of theobromine and caffeine, ABTS, and DPPH were purchased from Sigma-Aldrich.

### 2.2. Collection Location

This research was conducted in Mocajuba City, in Tauaré Island (latitude 2°37′58.130″ S and longitude 49°36′49.747″ W). The location is considered a floodplain region because it is located at the Tocantins River, named the *Baixo Tocantins* macroregion in Pará state, Brazil (Figure 1).

### 2.3. Fermentation and Drying Treatments and Sampling

Cocoa fruits (*Forastero* variety) were collected from Tauaré Island (Mocajuba city), manually opened with stainless knives, and the seeds with pulp were extracted in the fermentation process. Over six days, four types of different boxes were used: (a) a hexagonal box (PHB) measuring 0.4 m (height) × 0.6 m (width) × 0.3 m (length); (b) a square box (SB) measuring 1.0 m (height) × 1.0 m (width) × 1.20 m (length); (c) the HP is a handcrafted structure made with woven plant fibers. It creates a robust and ventilated basket with a capacity of 30 kg of cocoa seeds. The basket is cylindrical and measures 0.7 m (height) × 0.6 m (diameter). Its interlaced walls are supported by vertical rods that stabilize and raise the base off the ground; and d) a local square box (LSB) measuring 1.0 m (height) × 1.0 m (width) × 1.20 m (length), involved in fermenting the seeds for 48 h.

The drying process was carried out in a solar oven at the fermentation site for six days (moisture content around 6.0%). The fermented cocoa beans from the LSB treatment were dried for seven days under natural sunlight (following the producer’s procedures).

Subsequently, all fermented and dried cocoa beans were sent to the laboratory for analysis.

### 2.4. Methods of Analysis

#### 2.4.1. Physicochemical Analysis of Fermented and Dried Cocoa Beans

The husks and the embryos of the fermented and dried cocoa beans were manually separated, and the cotyledons were milled using an analytical mill (model A11, Ika, Staufen, Germany). The physicochemical analyses (the content of moisture, lipids, ashes, carbohydrates, pH, and titratable acidity—TTA) were carried out, according to the Association of Official Analytical Chemists—AOAC [24].

#### 2.4.2. Methylxanthines and Phenolic Compounds

Fermented and dried cocoa beans in all the treatments were previously defatted with n-hexane, according to Brito et al. [25] procedures. For the preparation of extracts, 1 g of defatted cocoa bean samples were mixed with an ethanol/distilled water solution (1:1, *v*/*v*) using a vortex for 5 min in conical tubes. The tubes were subjected to an ultrasonic bath (Eco-Sonics, Q3.0L model, São Paulo, SP, Brazil) at room temperature (~25 °C) for 10 min, with a fixed frequency of 25 KHz and a power of 120 w. All the extracts were filtered using 0.22 μm nylon membranes (Analítica, São Paulo, SP, Brazil).

The colorimetric method, based on Folin–Ciocalteu, was carried out to determine the total phenolic compounds, using catechin as the standard for the analytical curve (R^2^ > 0.99) [19]. The results were expressed in milligram equivalent epicatechin per gram sample (mg ECE/g).

The identification and quantification of methylxanthines (theobromine, theophylline, and caffeine), as well as phenolic compounds, including catechin, procyanidin B12, and epicatechin, were performed using an UHPLC system (Thermo Scientific Ultimate 3000, San José, CA, USA). This system is equipped with a quaternary pump (LPG-3400RS) and follows the methodology described by Rodríguez-Carrasco et al. [26].

Isocratic elution was performed using ultrapure water (component A) and acetonitrile (component B), both acidified with 2.5% acetic acid, which were filtered through a vacuum filtration system using 0.22 µm nylon membranes (Allcrom, São Paulo, Brazil). The flow rate was set at 0.2 mL/min, and the isocratic method was maintained at 7% acetonitrile for 10 min. Separation was achieved using a C18 column (Kinetex EVO, 100 Å, 1.7 µm, 100 × 2.1 mm) from Phenomenex (Torrance, CA, USA).

The extracts were carefully filtered through a 0.22 μm nylon membrane (Analítica, São Paulo, Brazil) and injected in a volume of 5 μL. Detection was conducted within a wavelength range of 200 to 515 nm, with 280 nm specifically used for quantifying alkaloids and phenolic compounds (catechin and procyanidin). The retention times and spectral data obtained were compared to individual standards for accurate compound identification. Quantification was performed using a high-precision calibration curve (R^2^ > 0.99), expressing the results in mg/g.

#### 2.4.3. Total Flavonoids

The total flavonoid content was measured using the method outlined by Fagundes et al. [27]. An aliquot of 250 μL of the extract was combined with 2.7 mL of ethanol and 120 μL of 0.5 M sodium nitrate (NaNO_3_). This mixture was homogenized with 120 μL of 0.3 M aluminum chloride (AlCl_3_) and allowed to stand for 10 min. Finally, the solution’s absorbance was measured using an UV–visible spectrophotometer (EVO 60, Thermo Fisher Scientific, Waltham, MA, USA) at 510 nm. The results were expressed in milligram equivalent catechin per gram sample (mg CAT/g).

#### 2.4.4. Antioxidant Capacity of Fermented and Dried Cocoa Beans

##### DPPH

To prepare the extracts, 1 g of fermented and dried cocoa was weighed and homogenized with 20 mL of ethanol, using a Turratec crusher (Tecnal TE-102, Piracicaba, Brazil), and subjected to ultrasonic treatment for 5 min. Afterward, the sample was centrifuged at 10,000 rpm for 10 min. The resulting supernatant was filtered and diluted to a final volume of 10 mL with ethanol.

The antioxidant activity using the DPPH method was performed according to Balcázar-Zumaeta et al. [28], based on the DPPH radical reduction capability (2,2-Diphenyl-1-45-Picrylhydrazyl). For this purpose, 100 μL of cocoa extract and 3.9 mL of DPPH solution were used. Absorbances were measured using an UV–visible spectrophotometer (EVO 60, Thermo Fisher Scientific, Waltham, MA, USA) at 517 nm. A standard curve was performed using Trolox (R^2^ > 0.99), expressing the results in µmol TE/g sample.

##### ABTS

This method is based on the free radical scavenging activity of ABTS (2,2-azino-bis(3-ethylbenzothiazoline-6-sulfonic acid) [28]. The ABTS+ radical was generated by reacting a 7 mM ABTS solution with a 140 mM potassium persulfate solution, followed by incubation in the dark at 25 °C for 16 h. After formation, the radical solution was diluted in 95% ethanol, until it reached an absorbance of 0.700 at 734 nm. Subsequently, an aliquot of 30 µL of the extracts was transferred, under dark conditions, to test tubes containing 3 mL of the ABTS+ radical solution. The absorbance at 734 nm was measured using an UV–visible spectrophotometer (EVO 60, Thermo Fisher Scientific, Waltham, MA, USA). A standard curve was performed using Trolox (R^2^ > 0.99) to determine the antioxidant activity, and the results were expressed in µmol TE/g sample.

##### FRAP

The FRAP reagent was prepared by initially formulating the TPTZ solution. Hydrochloric acid (HCl) was prepared at a concentration of 0.4 M by taking 166 µL of 37% HCl and diluting it in 50 mL of ultrapure water. Subsequently, 31.2 mg of TPTZ were weighed and dissolved in 10 mL of the 0.4 M HCl solution. For the measurement, the working solution was prepared by mixing 10 mL of acetate buffer (pH 3.6), 1 mL of the previously prepared TPTZ solution, and 1 mL of ferric chloride hexahydrate (0.2 M). The mixture was then subjected to the addition of the appropriate sample extract. The reaction was then carried out in a water bath (Indumelab, BM-10, Lima, Peru) at 30 °C for 4 min. The resulting mixture’s optical density was measured at a wavelength of 593 nanometers using a UV-Vis spectrophotometer (EMCLAB, EMC-11-UV, Duisburg, Germany) [28]. To ensure the accuracy of the results, a standard curve was generated using Fe^2+^, ensuring a coefficient of determination (R^2^) greater than 0.99. The results were expressed in μmol Fe^2+^/100 g of sample.

#### 2.4.5. Chemical Elements (MEC)

The partial chemical composition of the minerals was determined using an acid digestion process with 65% HNO_3_ and HCl. The mixture was allowed to rest for 17 h before being heated in a digestion block at 90 °C for one hour. After cooling to room temperature, the solution was diluted to a final volume of 45 mL with Milli-Q water. The mineral content was then analyzed using atomic absorption spectrometry (AAS) with an Ice 3000 Series AA spectrometer [29]. The results were expressed in μg/g.

### 2.5. Statistical Analysis

The results were compared with ANOVA and mean values were compared by Tukey’s test (*p* < 0.05).

Principal Component Analysis (PCA) was performed to reduce dimensionality and enhance experimental data visualization related to nutrient distribution in fermentation containers. This approach allowed for an assessment of the process efficiency based on fermented and dried cocoa bean quality. The preprocessing step involved autoscaling to standardize the models. A *p*-value of less than 0.05 was considered statistically significant.

Spearman’s non-parametric correlation examined the relationship between soil nutrients and the nutrients accumulated in cocoa beans after fermentation. The Spearman correlation coefficient ranges from −1 to 1 and was calculated with a 95% confidence level.

All the statistical analyses were performed using MINITAB software (LLC, State College, PA, USA), version 14.13.

## 3. Results and Discussion

### 3.1. Physicochemical Analysis of Cocoa Beans

As presented in Table 1, the PHB, SB, and HP samples exhibited moisture contents of 5.37%, 5.24%, and 5.06%, respectively. The LSB sample had a higher moisture content (6.16%) compared to the other samples. Similarly, Leite et al. [30] observed a moisture content of 6.95%, which exceeded the average and was close to the maximum value recorded for the LSB sample.

Fermented and dried cocoa beans must meet specific quality standards to ensure consistency throughout processing. To maintain optimal preservation and safety, moisture levels exceeding 7.5% serve as preliminary indicators of potential quality deterioration [30]. Other key factors influencing moisture reduction are cocoa bean size and the drying method, both of which affect water removal efficiency. Due to their larger surface area relative to volume, they facilitate more effective moisture evaporation [31].

The pH values exhibited variations, ranging from 4.65 to 5.78. The highest pH was observed in the HP sample, while the lowest was recorded in the PHB sample. The SB sample had a pH of 5.75, whereas the LSB sample exhibited the lowest value at 5.07, as shown in Table 1.

The natural microbiological process of fermentation occurs in two distinct stages, as proposed by De Vuyst and Leroy et al. [32], in which during the first stage (anaerobic phase), favorable pH and sugar contents can promote the development of yeasts and lactic and acid bacteria within 48 h [32,33]. The second stage corresponds to the active fermentation phase. The results suggest that the lower pH observed in the PHB sample can be attributed to the hexagonal format of the fermentation boxes, which made it difficult to maintain uniformity in the arrangement of the cocoa beans during fermentation and, consequently, during the stirring stage. Some beans remained static at the ends of the PHB. The minimum pH and maximum TTA values suggest that there was no adequate evaporation of the acids formed during fermentation [15].

The LSB value serves as a key indicator of the fermentation period, representing the completion of the first cycle under favorable microbiological conditions. Although slightly lower than the ideal threshold, further biochemical interactions are unlikely to significantly alter this value. Gaspar et al. [19] reported pH values of 4.07 and 4.09, which were slightly lower than those observed in the PHB samples. In contrast, Efraim et al. [34] documented a higher pH of 4.73%. In a subsequent analysis, the same authors found that cocoa beans subjected to natural drying exhibited an even greater pH increase, reaching 5.84%. Similarly, Leite et al. [30] recorded a pH of 5.20, while Ferreira et al. [13] reported values ranging from 3.75 to 5.24 in dried cocoa beans. These findings highlight the variability in pH levels among different fermentation and drying conditions, emphasizing the influence of post-harvest processing on cocoa bean chemistry.

Calvo et al. [11] further supported this correlation, highlighting that cacao beans with a pH below 4.5 tend to exhibit lower flavor potential, whereas those with a pH above 5.0 are associated with enhanced complexity and intensity in the final chocolate product. This pH distinction influences the development of key flavor precursors, potentially leading to chocolates with soft sensory attributes when acidity remains too high. In this context, the results obtained from fermented and dried samples across different fermentation boxes, such as SB, HP, and LSB, are promising. The observed pH values suggest optimal conditions for flavor development, reinforcing the potential for producing chocolates with more pronounced and complex sensory profiles.

The variation in titratable acidity across samples reflects its role as a key determinant of cocoa bean quality. The PHB sample exhibited the highest acidity (27.09 meq NaOH/100 g), followed by the LSB sample (21.27 meq NaOH/100 g). In contrast, the HP sample presented the lowest acidity (12.08 meq NaOH/100 g), while the SB sample showed an intermediate value of 14.70 meq NaOH/100 g.

Acidity levels in cocoa beans are influenced by both fermentation and drying processes, which regulate acid retention and volatilization. The optimal acidity for industrial applications falls within the range of 12 to 15 meq NaOH per 100 g, with partial seed drying before fermentation serving as an effective strategy to reduce acidity [11]. This reduction can be achieved through artificial drying methods or direct sun exposure [35]. Furthermore, natural drying plays a significant role in decreasing both volatile and total acidity, primarily due to the evaporation of acetic acid. This process promotes acid volatilization, leading to a gradual decline in acidity and ultimately shaping the sensory attributes of the beans [36]. The observed changes across different drying methods highlight the importance of post-harvest management in optimizing cocoa bean chemistry and flavor development [11,31].

The drying process is one of the primary factors influencing the acidity of fermented cocoa beans. Schwan et al. [37] mentioned that fermentative microorganisms contribute to acidity reduction by metabolizing citric acid present in the pulp. Additionally, the practice of stirring during fermentation plays a critical role in regulating temperature, which can rise to 45–51 °C. This controlled temperature increase promotes the proliferation of acetic and lactic acid bacteria, which, in turn, significantly influence subsequent enzymatic activity [32]. These enzymatic reactions are fundamental for the development of key flavor and aroma precursors in fermented cocoa beans.

Regarding ash content, all four samples showed levels close to 2%: PHB at 2.32%, SB at 2.44%, HP at 2.97%, and LSB at 2.34%. These ash contents were statistically different and slightly higher than the PHB sample. These results align with studies like that of Efraim et al. [34], which demonstrated that cocoa beans not subjected to fermentation had a higher ash content. In contrast, the ash content decreased after the samples underwent seven days of fermentation and drying (Table 1).

Hu et al. [38] reported an ash content of 2.8% in their study, which aligns with our findings (Table 1). Similarly, Efraim et al. [34] observed slightly higher values, measuring 3.21%. These findings reinforce the evidence that fermentation can reduce the ash content of the samples, supporting our observations.

The analyzed samples exhibited variations in quality parameters, as defined by the Cocoa Index [39]. In the PHB and LSB treatments, the total titratable acidity (TTA) exceeded the established limit of 20 meq NaOH 0.1 N/100 g, resulting in pH values outside the acceptable range for this parameter (5.60–6.57). Conversely, the samples from the SB and HP treatments remained within the quality limits specified by the index.

This consistency with the previously published literature suggests that no excessive ash levels were detected that could indicate impurities or contaminants negatively affecting the final product’s color, taste, or texture.

The analysis of different types of fermentation boxes (PHB, SB, HP, and LSB) revealed protein concentrations ranging from 12.89% to 15.00%. The SB sample exhibited the highest protein content (15.00%), indicating that fermentation may enhance protein retention in the samples. In contrast, the HP sample presented the lowest concentration (12.89%). The PHB and LSB samples showed intermediate protein levels of 14.71% and 14.38%, respectively.

Studies conducted by Hu et al. [38] found similar results to those of the PHB and LSB samples, revealing a percentage of 14.3%. Efraim et al. [34] reported data showing values ranging from 14% to 17% in their fermented and dried samples. Additionally, the authors found a value of 16.99% for fermented cocoa beans that underwent fermentation for seven days followed by natural drying.

Cocoa fermentation comprises two distinct biochemical processes: the anaerobic metabolism of sugars and the hydrolysis of proteins, both of which are crucial for the development of the final product’s sensory attributes. The anaerobic metabolism of sugars occurs during the initial phase of fermentation and is driven by yeasts and lactic acid bacteria present in the cocoa pulp [15]. These microorganisms metabolize the available carbohydrates, producing ethanol, organic acids, and other metabolites that alter the pH and chemical composition of the environment. This microbial activity facilitates the transition from anaerobic to aerobic conditions, enabling subsequent biochemical transformations essential for cocoa quality [15,32,40].

Concurrently, within the technological framework of fermentation, the hydrolysis of proteins stored in the cocoa beans cotyledons takes place. This process involves proteolytic enzymes that break down proteins into free amino acids and small peptides.

During the aerobic phase of cocoa fermentation, acetic acid bacteria, primarily species from the *Acetobacter* genus, play a crucial role by oxidizing ethanol into acetic acid through an exothermic reaction [15]. This process leads to embryo inactivation, increased cell wall permeability, and the release of precursor molecules responsible for chocolate aroma and flavor development. The associated temperature rise, often exceeding 40 °C, promotes the enzymatic degradation of polyphenolic compounds via polyphenol oxidase activation. Concurrently, methylxanthines, such as theobromine, caffeine, and theophylline, are partially degraded and exuded from the cocoa beans during mass turning. The monitoring of the compound content of these groups of compounds is essential for quality control, as these compounds contribute to bitterness and astringency. However, in moderate concentrations, they exhibit physiological benefits, including stimulant, vasodilatory, diuretic, and muscle relaxant properties [15,32].

A survey conducted by Hu et al. [38] indicated that various factors, including fermentation and fruit ripening, influence the chemical composition of fermented cocoa beans. González et al. [17] elaborated further, noting that protein content may decrease during fermentation. Lima et al. [41] highlighted that although cocoa beans have a high protein content during their development, this amount remains constant through the processing stages despite hydrolysis and other biochemical reactions.

The literature estimates a protein percentage of up to 17% for fermented and dried cocoa beans [39,42]. Data may vary due to factors that influence fermentation conditions, such as the efficiency of hydrolytic phases, the number of amino acids, and the precursors of enzymes responsible for color and flavor. The protein content results showed a significant difference only in the lower value of the HP sample, likely due to a shorter fermentation time, which resulted in lower microbiological activity.

The present study determined lipid levels of 49.8% (HP), 41.5% (PHB), 46.6% (SB), and 37.51% (LSB). These findings are critical for assessing the quality and stability of cocoa-derived products, such as chocolate. Almond fat not only plays a fundamental role in product preservation but also possesses significant health benefits. The lipids in the samples exhibit antioxidant and anti-inflammatory properties, which have been shown to contribute to human health, as highlighted by Melo et al. [43]. These results emphasize the broader implications of our research in food science and nutrition.

The high lipid content typically indicates the maintenance of fat in the cotyledons without migration to the peel during drying, which is usually discarded, as described by Chagas Junior [42]. According to Chen et al. [44], the proportion of fatty acids can affect the physical and sensory characteristics of cocoa products, such as fat content and melting point.

The lipid levels found in fermented cocoa beans from different types of fermentation boxes fall within the expected range, according to the literature [39], indicating satisfactory quality. Notably, the HP and PHB samples exhibited higher lipid values, with the highest measured at 49.8% in the HP sample. In contrast, the SB and LSB samples showed lower lipid levels, with the LSB sample recording the lowest value at 37.51%. Several factors can influence the amount of lipids in cocoa beans, including variety, storage conditions, and the drying process. This research identifies the fermentation method as a key indicator of lipid quality.

The carbohydrate content in fermented and dried cocoa beans ranged between 29.28% and 39.61%. The HP sample obtained the lowest value (29.28%), while the LSB sample had the highest (39.61%). The other samples, SB and PHB, presented values of 30.72% and 36.1%, respectively. The HP bed, characterized by its less compact structure and higher ventilation index, generally contains a lower total carbohydrate content than the LSB bed. This difference can be attributed to the reduced availability of sugars in the HP bed, which affects the other compounds that serve as substrates for fermentation in this environment [15].

### 3.2. Analysis of Total Phenolic Compounds and Total Alkaloids

Total polyphenols are antioxidant compounds found in various foods, including cocoa (Table 2). Our research revealed significant variability in the total polyphenol content of different samples of dried cocoa beans (PHB, SB, HP, and LSB), with values ranging from 9.72 to 28.79 milligram equivalent epicatechin per gram sample (mg ECE/g). The PHB sample had the highest total polyphenol content (28.79 mg ECE/g), followed closely by the HP sample (26.91 mg ECE/g). In contrast, the LSB sample exhibited the lowest concentration of total polyphenols (9.72 mg ECE/g). The SB sample had an intermediate value of 19.40 mg ECE/g.

Efraim et al. [34] reported lower phenolic compound levels, ranging from 90.38 to 102.12 mg/100 g. Their study investigated the potential degradation of phenolic compounds during a seven-day fermentation period, followed by natural and artificial drying methods. The findings revealed that oven drying led to an 11.6% reduction in phenolic content, whereas sun drying resulted in a significantly lower loss of only 2.8%. These results underscore the advantages of natural drying, particularly sun exposure, in preserving phenolic compounds.

The number of polyphenols in cocoa decreases during fermentation and drying due to complex enzymatic reactions that oxidize polyphenols. Additionally, the drying temperature influences the degradation of phenolic compounds, reducing cocoa bitterness [40].

Calvo et al. [11] explained that differences in the levels of phenolic compounds must have many factors to be taken into consideration, such as genetic factors of the plant, fermentation, and drying methods. Niemenak et al. [45] highlighted another important factor: the abiotic stress experienced by the plant due to low water supply, high solar incidence, and low humidity. These conditions directly impact the production of phenolic compounds.

The samples analyzed demonstrate levels of phenolic compounds are the same compared to existing literature [46]. This enhancement can be attributed to intermittent cultivation in floodplain areas, leading to fruits with a greater concentration of bioactive compounds during maturation. The PHB and HP fermentation methods help minimize losses. However, it is essential to evaluate whether the increased levels of phenolic compounds lead to heightened astringency, potentially impacting the sensory characteristics of the fermented cocoa beans.

Theobromine content ranged from 0.013 to 0.0225 mg/100 g, with the LSB sample presenting the lowest value (0.013 mg/100 g) and the PHB sample the highest (0.0225 mg/100 g). The SB sample yielded 0.024 mg/100 g, while the HP sample showed 0.030 mg/100 g. Regarding theophylline, no results were observed for the PHB sample, whereas the SB, HP, and LSB samples exhibited similar minimum values.

In the analysis of caffeine levels, the samples displayed the following concentrations: 0.8 mg/g in PHB, 1.0 mg/g in SB, 0.4 mg/g in PH, and 0.2 mg/g in LSB. These findings are lower than those reported by Collazos-Escobar et al. [47], who analyzed fermented and dried cocoa beans from the *Trinitario* and *Forastero* varieties and found higher caffeine levels. Their study reported theobromine values ranging from 18.7 mg/g to 20.6 mg/g, while caffeine values varied between 1.7 mg/g and 2.9 mg/g. Gaspar et al. [19] also reported theobromine concentrations of 8.37 mg/g and 8.50 mg/g, along with caffeine concentrations of 2.45 mg/g and 2.55 mg/g.

The results from our study on theobromine, theophylline, and caffeine are consistent with those of Gaspar et al. [19], who observed that theophylline levels in cocoa beans are either negligible or absent, which aligns with our observations. Additionally, our findings support Gaspar’s assertion that theobromine is the primary methylxanthine in cocoa beans, typically found in higher concentrations. Furthermore, theobromine serves as a precursor to the caffeine biosynthetic pathway, as noted by Martínez-Pinilla et al. [48].

Our analysis of catechin levels in the samples revealed values ranging from 0.2 mg/g to 0.4 mg/g. The LSB sample exhibited the lowest catechin content (0.2 mg/g), while the HP sample had the highest (0.4 mg/g). The PHB and SB samples showed intermediate values of approximately 0.3 mg/g. Maintaining these catechin levels is crucial for optimizing the balance of raw materials in the industry, particularly in relation to sensory and aromatic properties as well as health benefits [39].

Gaspar et al. [19] reported catechin concentrations between 0.37 and 0.41 mg/g, while Carrillo et al. [49] found values ranging from 0.226 to 1.297 mg/g in fermented and dried cocoa beans. Procyanidin levels varied from 0.6 to 0.8 mg/g, with HP and LSB samples showing the highest concentration (0.8 mg/g). The PHB sample had a value of 0.6 mg/g, while the SB sample exhibited 0.7 mg/g.

The effects of different drying methods, specifically solar and artificial drying, on procyanidin concentrations in cocoa beans suggest that sun drying may be more effective in preserving procyanidins in fermented cocoa beans, even though it initially causes greater degradation of monomers. Factors such as temperature, solar radiation, and enzyme activity significantly influence the formation and degradation of procyanidins during the drying process [50].

Table 3 summarizes the concentration ranges of the flavonoids and the antioxidant capacity assessment methods employed to evaluate fermented and dried cocoa beans across different fermentation box types (PHB, SB, HP, and LSB; see Table 3 for bed specifications). The antioxidant assays (e.g., DPPH, FRAP, and ORAC) highlight variations in oxidative activity, correlating with the residual flavonoid profiles post-fermentation.

### 3.3. Antioxidant Activity of Fermented and Dried Cocoa Beans

Analysis of total flavonoids in the PHB, SB, HP, and LSB samples revealed that the SB sample had the lowest content, approximately 16.68 mg CAT/g. The PHB sample contained 17.41 mg CAT/g. In contrast, the HP sample exhibited the highest value, 35.14 mg CAT/g, while the LSB sample showed an intermediate value of 24.72 mg CAT/100 g (Table 3).

Efraim et al. [34] reported flavonoid concentrations in dried almond samples ranging from 86.75 mg CAT/g to 148.70 mg CAT/g. These findings highlight the impact of the drying process on flavonoid preservation and suggest that high temperatures can lead to their degradation. However, the authors emphasized that the fermentation stage is the primary cause of flavonoid loss. Flavonoid levels can also vary based on factors such as cocoa surface characteristics, geographical origin, maturity level, harvest time, post-harvest conditions, fermentation, drying processes, and storage conditions [38].

Fermentation plays a crucial role in minimizing flavonoid loss, highlighting the importance of knowing and controlling this process. High-pressure (HP) methods demonstrated better preservation of total flavonoids, suggesting that the basket structure limits the fermentation process, thereby reducing oxidative and enzymatic reactions. Consequently, seeds fermented using this method tend to develop a more intense flavor with prominent astringent notes [51].

Analysis using this method yielded values ranging from 60.8 to 67.80 μM Trolox equivalents/g, with distinct variations observed among samples. The PHB bed exhibited the lowest antioxidant capacity (60.8 μM Trolox/g), whereas the LSB bed demonstrated the highest activity (67.80 μM Trolox/g). Intermediate values were recorded for SB (16.68 μM Trolox/g) and HP (64.8 μM Trolox/g), consistent with their differential retention of bioactive compounds during processing.

The ABTS assay presented variable results, with the LSB sample exhibiting the lowest value (1.2 μM Trolox/g), followed by the SB sample (7.2 μM Trolox/g). The PHB sample reached 17.345 μM Trolox/g, while the HP sample showed the highest value (28.8 μM Trolox/g).

The low ABTS and FRAP values observed in LSB are likely due to its shorter fermentation time, which may have impacted the bioavailability of antioxidant compounds. In contrast, SB’s intermediate value suggests a moderate production of these compounds. Meanwhile, PHB displayed higher values, possibly due to its specific microbiota and fermentation conditions, indicating effective antioxidant compound production. Furthermore, HP’s broader range suggests superior antioxidant compound production, potentially linked to the fermentation method used.

Polyphenols, including flavonoids like catechins and epicatechins, are naturally occurring antioxidant compounds in cocoa. Enzymes and microorganisms partially break down these compounds during fermentation, reducing their overall content. However, this degradation can also produce new compounds with antioxidant properties, such as phenolic acids and other metabolites. The microorganisms involved in fermentation generate metabolites that may possess antioxidant capabilities, including various organic acids and phenolic compounds formed during the process, which can enhance the antioxidant capacity of fermented cocoa beans [19,46].

The varying results among the samples underscore the impact of several factors, including pH, ambient temperature, and fermentation time. These elements significantly influence the production of antioxidant compounds, emphasizing the complexity of this research and the need for further investigation.

### 3.4. Essential and Non-Essential Minerals

Table 4 presents the mean values of the essential and non-essential elements in dried cocoa beans from the PHB, SB, HP, and LSB fermentation boxes.

The concentrations of potassium and magnesium in the samples (7505.02 μg/g and 2668.13 μg/g, respectively) are in greater quantities. Potassium concentrations ranged from 1200 to 14,500 ppm in cocoa beans of various global origins [52], while Perea-Villamil et al. [53] observed values between 10,000 and 13,000 ppm in Colombian cocoa beans.

The variation in calcium content may be attributed to factors such as soil composition and climate [52]. Currently, there is no official comparative data regarding magnesium levels in cocoa beans. Calcium, an essential nutrient, plays a key role in biological functions, such as muscle contraction, blood clotting, nerve impulse transmission, and skeletal structural support [54,55]. The required calcium intake varies based on gender and age. The Dietary Reference Intakes (DRI) recommendations, established for the United States and Canada, are also used as a reference for the West African population, with an average age of 19 to 30 years. For both men and women, the adequate daily intake is 1000 mg [56].

Sodium was one of the nutrients that did not reach detectable values in any of the samples. The sodium content in other fermented dried bean samples ranged from 355.75 ppm to 600 ppm and 110 ppm.

Copper is crucial for bone growth and skeletal development, contributing to increased bone stiffness and maintaining bone quality. Additionally, copper functions as a catalyst in various oxidation processes in plants, highlighting its essential biological role [57]. In this research, the copper concentration range was from 26.13 μg/g (HP) to 33.35 μg/g (PHB). This element plays an essential role in the quality and safety of cocoa beans, influencing both of chemical composition and sensory aspects of the final product. During fermentation and drying, this micronutrient participates in enzymatic reactions that affect the oxidation of phenolic compounds, directly impacting the formation of color and flavor. In adequate concentrations, copper contributes to the activity of oxidative enzymes, while excess copper can intensify undesirable reactions, resulting in metallic flavors and bitterness in chocolate [58,59].

Manganese concentrations ranged from 13.09 to 19.61 μg/g, with the highest levels observed in the HP sample and the lowest in LSB. Beyond its quantitative presence, manganese plays a vital role in plant metabolism, particularly in photosynthesis, carbohydrate metabolism, and chlorophyll biosynthesis. As a key enzymatic cofactor, it supports respiration and nitrogen metabolism by activating oxidoreductases, decarboxylases, hydrolases, and phosphate group transferases. Additionally, manganese contributes to essential biosynthetic pathways, including protein synthesis and ascorbic acid formation, reinforcing its physiological significance [60].

Strontium concentrations were consistent across samples, showing no significant statistical differences. The highest strontium value was found in the HP sample at 13.17 μg/g. Similar values were observed in other samples, including SB at 12.48 μg/g, LSB at 12.80 μg/g, and PHB at 12.95 μg/g. Vanadium concentrations varied significantly, with PH measuring 81.67 μg/g, SB at 28.46 μg/g, LSB at 45.53 μg/g, and PHB at 51.17 μg/g. No comparative data were found for strontium and vanadium levels in dried cocoa beans.

Regarding aluminum, detectable values were found only in PHB (61.37 μg/g) and LSB (53.62 μg/g), while the SB and PHB samples did not show measurable concentrations. The World Health Organization recommends a maximum weekly aluminum intake of 1 mg/kg body weight. Aluminum is a non-essential element for humans, and its adverse effects are primarily associated with neurological disorders, including Alzheimer’s and Parkinson’s diseases [61,62] According to the available literature, no data were collected on barium content in cocoa beans.

Although minerals such as potassium (K), calcium (Ca), magnesium (Mg), sodium (Na), iron (Fe), zinc (Zn), copper (Cu), manganese (Mn), strontium (Sr), vanadium (Vn), aluminum (Al), and barium (Ba) are naturally present in cocoa beans and have recognized roles in plant metabolism and microbial physiology, their specific functions during cocoa fermentation remain poorly understood. To date, there is limited and fragmented information regarding how these microelements influence microbial succession, enzymatic activity, or the chemical transformations that occur during the fermentation process.

The lack of targeted studies exploring the dynamic interactions between mineral availability, microbial metabolism, and metabolite production in cocoa pulp and natural seeds, represents a critical knowledge gap. Therefore, the precise contribution of these elements to fermentation kinetics, flavor precursor development, and fermented cocoa quality has yet to be clearly defined and warrants further investigation.

The results in Table 1, Table 2, Table 3 and Table 4 reveal significant differences in the physicochemical parameters, phenolic compounds, and antioxidant capacity of cocoa beans fermented and dried in different fermentation boxes (PHB, SB, HP, and LSB). Although the design of the boxes was initially considered the main factor influencing these variations, it is essential to recognize that the microbial communities responsible for fermentation play a central role in mediating these differences.

Cocoa fermentation is a complex microbiological process in which yeasts, lactic acid bacteria, and acetic acid bacteria act sequentially to transform pulp sugars into organic acids, alcohols, and aromatic compounds, directly affecting the chemical and nutritional composition of the beans [37]. Although the study did not directly analyze the microbiota, the results indicate that the design of the boxes influences the fermentation conditions (e.g., aeration or time), shaping the microbial succession described by Vuyst and Leroy [32]. Future studies could clarify the species involved and their contribution to the observed profiles, complementing this initial chemical analysis.

### 3.5. PCA: Analysis of Minerals in Different Fermentation Boxes

#### Self-Analysis of the Covariance Matrix

To analyze the similarities and differences among the four types of fermentation boxes used in cocoa processing, a Principal Component Analysis (PCA) was performed. The first two principal component (PC1 + PC2) axes accounted for 99.9% of the total data variation, as illustrated in the upper part of Figure 2. The HP bed contributed the most to forming the first axis (58.9%), showing a positive relationship. For the second principal component, the LSB bed was the primary contributor (81.9%), exhibiting a positive relationship (Table 5). These results highlight the critical role of fermentation bed characteristics in shaping the final product’s quality.

Rahardjo et al. [63] performed a similar analysis to evaluate how different controlled fermentation systems—specifically, jacket systems, sun drying, and wooden boxes—affect the quality and acceptability of cocoa beans. Their use of Principal Component Analysis revealed significant differences in the chemical composition of volatile compounds, which were primarily associated with variations in the pulping process rather than the type of controlled fermentation system used.

Regarding the relevance of nutrients in the fermentation boxes, Principal Component Analysis (PCA) on the first two axes accounted for 99.9% of the total data variation. This high percentage (PC1: 89.1% and PC2: 10.8%) indicated that potassium (K) and magnesium (Mg) were the most substantial contributors to the overall variability. Potassium (K) contributed the most to forming the first axis (97.4%), showing a positive relationship. For the second axis, magnesium (Mg) was the primary contributor (96.7%), also exhibiting a positive relationship (Table 6). The PCA further revealed that the HP bed exhibited the highest potassium (K) concentration, loading 0.974 on PC1. Conversely, magnesium (Mg) showed a loading of 0.967, with concentrations of 7505.02 and 2668.13 μg/g, respectively.

The loss of minerals during fermentation, caused by leaching from mucilage in an acidic and highly soluble environment, presents an interesting opportunity for future research. The lower acidity observed in HP boxes, along with the higher levels of potassium (K) and magnesium (Mg) found in these fermentation boxes, indicates a possible connection between the type of fermentation bed used and the retention of essential nutrients (Figure 2).

## 4. Conclusions

This study evaluated, for the first time, the fermentation of cocoa beans from Tauaré Island’s floodplains in the Brazilian Amazon using different fermentation boxes. Differences were observed in the beans’ composition, phenolic compounds, and antioxidant capacity. The basket fermentation box (HP) showed good results in composition and antioxidants but may result in overly astringent fermented cocoa beans due to excess polyphenols. The local square box (LSB) promoted higher nutrient levels despite a short fermentation period, indicating efficient microbial activity. The hexagonal box (PHB) showed potential for future studies, though lacking in microbial activity compared to other types of fermentation boxes. The square box (SB) provided good results for flavonoids, polyphenols, and minerals, but its underdeveloped microbial activity affected almond standardization. No significant barium or sodium levels were found, confirming the cocoa’s organic nature. This floodplain cocoa shows promise as a sustainable, organic product with higher market value than dry land cocoa, though more research is needed due to limited scientific data.

To further explore these findings, future studies should incorporate microbiological analyses, such as metagenomic sequencing, to characterize the specific microbial populations in each type of fermentation box and elucidate their influence on the observed nutritional and antioxidant profiles.

## Figures and Tables

**Figure 1 foods-14-01391-f001:**
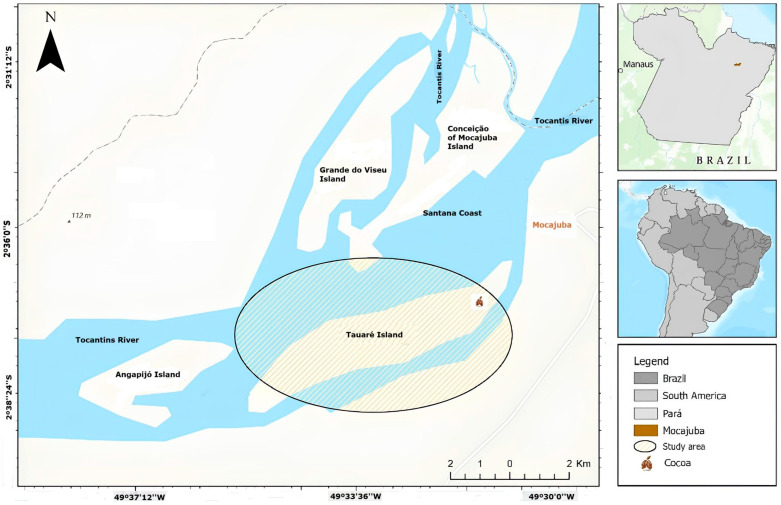
Geographic location of the study area in Mocajuba city, Pará State, Brazil.

**Figure 2 foods-14-01391-f002:**
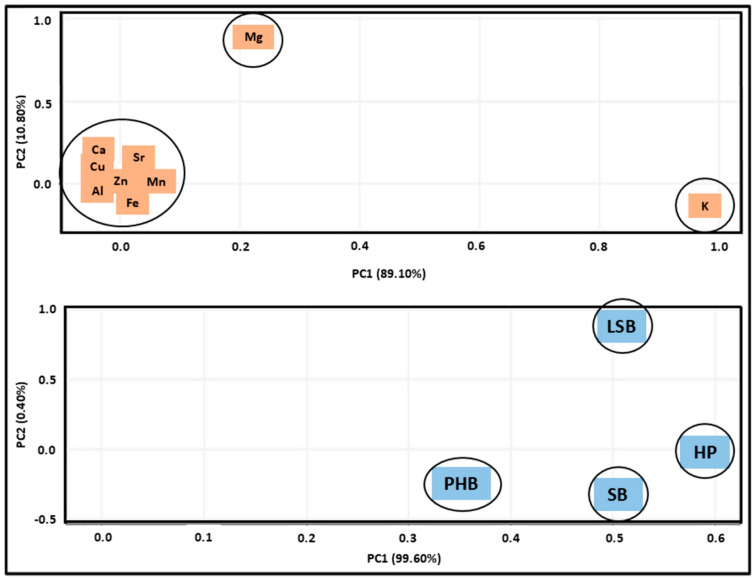
Principal Component Analysis (PCA) regarding nutrients and fermentation boxes.

**Table 1 foods-14-01391-t001:** Mean values * of physicochemical parameters of fermented and dried cocoa beans carried out in different types of boxes **, Mocajuba, Brazil, 2024.

Parameter	PHB	SB	HP	LSB
Moisture (%)	5.37 ± 0.09 ^b^	5.24 ± 0.08 ^bc^	5.06 ± 0.14 ^c^	6.16 ± 0.02 ^a^
pH	4.65 ± 0.08 ^c^	5.75 ± 0.07 ^ab^	5.78 ± 0.02 ^ba^	5.07 ± 0.20 ^b^
Titratable acidity—TTA ***	27.09 ± 0.8 ^a^	14.70 ± 1.1 ^c^	12.08 ± 0.96 ^d^	21.27 ± 0.8 ^b^
Ash (%)	2.32 ± 0.04 ^b^	2.44 ± 0.07 ^b^	2.97 ± 0.01 ^a^	2.34 ± 0.04 ^b^
Protein (%)	14.71 ± 0.25 ^a^	15.00 ± 0.15 ^a^	12.89 ± 0.15 ^b^	14.38 ± 0.54 ^a^
Lipids (%)	41.5 ± 0.18 ^c^	46.6 ± 0.03 ^b^	49.8 ± 0.16 ^a^	37.51 ± 0.41 ^d^
Carbohydrates (%)	36.1 ± 0.61 ^b^	30.72 ± 0.27 ^c^	29.28 ± 0.82 ^c^	39.61 ± 0.14 ^a^

* The means ± standard deviation with different letters on the same line are statistically different (Tukey’s test, *p* ≤ 0.05). ** PHB—hexagonal box, SB—square box, HP—traditional basket, LSB—local square box. *** meq. NaOH 0.1 N/100 g: milliequivalent sodium hydroxide solution 0.1 N per 100 g sample.

**Table 2 foods-14-01391-t002:** Means values * of total polyphenols and alkaloids of fermented and dried cocoa beans carried out in different types of boxes **, Mocajuba, Brazil, 2024.

Parameter	PHB	SB	HP	LSB
Total Polyphenols ***	28.79 ± 2.11 ^a^	19.40 ± 1.12 ^c^	26.91 ± 3.91 ^ab^	9.72 ± 1.55 ^d^
Theobromine (mg/g)	2.2 ± 0.2 ^c^	2.4 ± 0.2 ^b^	3.0 ± 0.2 ^a^	1.3 ± 0.1 ^d^
Theophylline (mg/g)	N/D ****	0.1 ± 0.00 ^a^	0.1 ± 0.01 ^a^	0.1 ± 0.01 ^a^
Caffeine (mg/g)	0.8 ± 0.00 ^c^	1.0 ± 0.1 ^b^	1.4 ± 0.1 ^a^	0.8 ± 0.01 ^c^
Catechin (mg/g)	0.3 ± 0.00 ^b^	0.3 ± 0.01 ^b^	0.4 ± 0.00 ^a^	0.2 ± 0.01 ^c^
Procyanidin (mg/g)	0.6 ± 0.00 ^c^	0.7 ± 0.01 ^b^	0.8 ± 0.00 ^a^	0.8 ± 0.01 ^a^

* The means ± standard deviation with different letters in the same line are statistically different (Tukey’s test, *p* ≤ 0.05). ** PHB—hexagonal box, SB—square box, HP—traditional basket, LSB—local square box. *** mg ECE/g: milligram equivalent epicatechin per gram sample. **** Theophylline was not detected. LOQ = 0.02 mg/g. N/D: not detected.

**Table 3 foods-14-01391-t003:** Means values * of total flavonoids, DPPH, ABTS, and FRAP of fermented and dried cocoa beans carried out in different types of boxes **, Mocajuba, Brazil, 2024.

Parameter	PHB	SB	HP	LSB
Total Flavonoids (mg CAT/g) ***	17.41 ± 1.99 ^ab^	16.68 ± 0.48 ^c^	35.14 ± 13.08 ^a^	24.72 ± 3.48 ^a^
DPPH (μM Trolox/g) ****	60.8 ± 1.12 ^a^	61.7 ± 4.78 ^a^	64.8 ± 4.8 ^a^	67.80 ± 0.29 ^a^
ABTS (μM Trolox/g) ****	17.345 ± 0.45 ^b^	7.2 ± 1.40 ^c^	28.8 ± 0.7 ^a^	1.2 ± 0.45 ^d^
FRAP (μmol Fe^2+^/100 g) *****	207.7 ± 31.0 ^b^	250.75 ± 2.1 ^b^	319 ± 3.0 ^a^	78.4 ± 12.7 ^c^

* The means ± standard deviation with different letters in the same line are statistically different (Tukey’s test, *p* ≤ 0.05). ** PHB—hexagonal box, SB—square box, HP—traditional basket, LSB—local square box. *** mg CAT/g: milligram equivalent catechin per gram sample. **** µM Trolox/g: micromole Trolox equivalent per gram sample. ***** μmol Fe^2+^/100 g: micromole of Iron per 100 g of sample.

**Table 4 foods-14-01391-t004:** Means values * of essential and non-essential elements of fermented and dried cocoa beans carried out in different types of boxes ** in μg/g, Mocajuba, Brazil, 2024.

Mineral	PHB	SB	HP	LSB
K	4720.64 ± 1259.50 ^b^	6352.64 ± 173.74 ^ab^	7505.02 ± 424.99 ^a^	6715.76 ± 753.06 ^ab^
Ca	489.66 ± 156.80 ^a^	512.53 ± 9.34 ^a^	391.66 ± 16.86 ^a^	363.2 ± 41.34 ^a^
Mg	1900.99 ± 517.52 ^bc^	2489.12 ± 31.43 ^ab^	2668.13 ± 110.63 ^a^	1643.13 ± 76.63 ^c^
Na	N/D	N/D	N/D	N/D
Fe	42.10 ± 5.19 ^b^	48.39 ± 8.67 ^a^	29.63 ± 1.8 ^c^	50.80 ± 3.89 ^a^
Zn	60.78 ± 7.69 ^b^	70.05 ± 9.35 ^a^	46.98 ± 5.65 ^c^	60.49 ± 3.48 ^b^
Cu	33.35 ± 4.84 ^a^	32.39 ± 2.53 ^a^	26.13 ± 1.36 ^b^	32.35 ± 1.39 ^a^
Mn	17.32 ± 2.23 ^b^	16.76 ± 0.92 ^b^	19.61 ± 1.47 ^a^	13.09 ± 0.91 ^c^
Sr	12.95 ± 1.5 ^a^	12.48 ± 0.24 ^a^	13.17 ± 1.67 ^a^	12.80 ± 1.12 ^a^
Vn	51.17 ± 8.57 ^b^	28.46 ± 5.54 ^d^	81.67 ± 4.07 ^a^	45.53 ± 4.74 ^c^
Al	61.37 ± 8.52 ^a^	N/D	N/D	53.62 ± 3.43 ^a^
Ba	N/D	N/D	N/D	N/D

* The means ± standard deviation with different letters in the same line are statistically different (Tukey’s test, *p* ≤ 0.05). ** PHB—hexagonal box, SB—square box, HP—traditional basket, LSB—local square box. N/D: not detected.

**Table 5 foods-14-01391-t005:** Principal Component Analysis (PCA) of fermented and dried cocoa beans carried out in different types of boxes: PHB—hexagonal box, SB—square box, HP—traditional basket, LSB—local square box, Mocajuba, Brazil, 2024.

	PHB	SB	HP	LSB
PC1	0.371	0.501	0.589	0.514
PC2	−0.374	−0.416	−0.125	0.819

**Table 6 foods-14-01391-t006:** Factor loadings of nutrients on main components (PC1 e PC2).

Mineral	PC1	PC2
K	0.974	−0.216
Mg	0.222	0.967
Cu	0.001	−0.005
Mn	0.001	0.004
Fe	0	−0.014
Sr	0	0.001
Zn	−0.002	−0.003
Ca	−0.036	0.121

## Data Availability

The original contributions presented in the study are included in the article, further inquiries can be directed to the corresponding author.

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
