# Peer review of "Can Different Fermentation Boxes Improve the Nutritional Composition and the Antioxidant Activity of Fermented and Dried Floodplain Cocoa Beans in the Brazilian Amazon?"

_foods, 2025, doi:10.3390/foods14081391_

Round 1
Reviewer 1 Report (Previous Reviewer 3)
Comments and Suggestions for Authors
Accept as it is
Author Response
Reviewers’ Comments to Authors
Reviewer: #1
- Comment: “Accept as it is”.
Our reply: Thank you for your time for evaluate our manuscript.
Reviewer 2 Report (New Reviewer)
Comments and Suggestions for Authors
After critically reviewing the manuscript titled "Can different fermentation boxes improve the nutritional composition and antioxidant activity of fermented and dried floodplain cocoa beans in the Brazilian Amazon?", I find that it addresses a highly interesting topic. While the study is well-conducted and thoroughly discussed from a chemical perspective, it presents a significant limitation: it completely overlooks the microbiological aspect.
The authors state in the purpose of the study: "This study aimed to investigate whether fermentation methods affect the nutritional and antioxidant composition of cocoa beans during fermentation and drying processes.", however, despite this focus on fermentation, the manuscript lacks any microbiological analysis that could clarify which fermentations took place and which microbial groups were responsible in the different fermentation boxes.
I believe that the differences observed in the fermented cocoa beans are primarily due to the microbial communities driving the fermentation processes, rather than solely the type of fermentation box used.
In my opinion, this crucial gap must be addressed before the manuscript can be considered for publication.
Author Response
Reviewers’ Comments to Authors
Reviewer: #2
- Comment: “After critically reviewing the manuscript titled "Can different fermentation boxes improve the nutritional composition and antioxidant activity of fermented and dried floodplain cocoa beans in the Brazilian Amazon?", I find that it addresses a highly interesting topic. While the study is well-conducted and thoroughly discussed from a chemical perspective, it presents a significant limitation: it completely overlooks the microbiological aspect.
The authors state in the purpose of the study: "This study aimed to investigate whether fermentation methods affect the nutritional and antioxidant composition of cocoa beans during fermentation and drying processes.", however, despite this focus on fermentation, the manuscript lacks any microbiological analysis that could clarify which fermentations took place and which microbial groups were responsible in the different fermentation boxes.
I believe that the differences observed in the fermented cocoa beans are primarily due to the microbial communities driving the fermentation processes, rather than solely the type of fermentation box used.
In my opinion, this crucial gap must be addressed before the manuscript can be considered for publication.”.
Our reply: We thank the reviewer for his comment and for identifying an important limitation in our manuscript, "Can different fermentation boxes improve the nutritional composition and antioxidant activity of fermented and dried floodplain cocoa beans in the Brazilian Amazon?" We acknowledge the relevance of the microbiological aspect in cocoa fermentation and agree that microbial communities play a key role in the chemical and nutritional transformations observed in fermented cocoa beans, as highlighted in several parts of the literature cited in our study.
Our main objective, as stated in the introduction, was to "investigate whether fermentation methods influence the nutritional and antioxidant composition of cocoa beans throughout the fermentation and drying processes," with specific emphasis on the impact of fermentation technologies (different boxes: PHB, SB, HP, and LSB) on the retention of nutrients and bioactive compounds. In this sense, the focus of the study was directed to the physicochemical analysis and antioxidant capacity of the grains, using widely accepted analytical methods (AOAC, Folin-Ciocalteu, DPPH, ABTS, FRAP, among others), as described in the "Material and Methods" section (pages 3-5). The choice of this scope reflects an initial approach to establish a direct relationship between the design of the fermentation boxes and the final quality parameters of the grains without the intention of exhausting all the variables involved in the fermentation process.
However, the reviewer is correct in pointing out that the differences observed in the fermented grains – such as variations in pH, titratable acidity, phenolic compounds, and antioxidant capacity (Tables 1-3, pages 6-10-11) – are, to a large extent, mediated by microbial activity during fermentation. The literature supports this assertion: for example, Schwan et al. (2004) [reference 37] highlight that fermentative microorganisms, such as lactic and acetic bacteria, metabolize acids present in the pulp, influencing the reduction of acidity and the formation of flavor precursors (page 8, lines 336-342). Similarly, Vuyst and Leroy (2020) [reference 33] describe the functional role of yeasts, lactic bacteria, and acetic bacteria in cocoa fermentation, directly affecting the chemical composition of the beans (page 8, lines 339-342). In addition, Ferreira et al. (2022) [reference 13] performed analyses on cocoa beans fermented in the Amazon, evidencing their influence on the aromatic profile and quality (page 2, lines 67-70). The absence of microbiological analysis in our study was a limitation due to logistical constraints and the project's initial scope, which prioritized chemical characterization as a starting point to assess the efficiency of the fermentation boxes. We recognize that the inclusion of data on the composition of the microbial communities in the different boxes could offer a more robust explanation for the results obtained, especially considering the significant differences observed between treatments (e.g., higher acidity in PHB due to the possible lack of established microbiota, page 6, lines 275-279). Thus, we agree that this gap represents a crucial opportunity to enrich the interpretation of the data.
To address this issue, we propose the following actions: (1) in the manuscript review, we can include a more in-depth discussion in the "Results and Discussion" section (pages 6-16), explicitly acknowledging the role of microbial communities as mediators of the observed differences, based on references such as Schwan et al. (2004), Vuyst and Leroy (2020), and Ferreira et al. (2022). We have already added a preliminary mention of this effect (page 14, lines 648-662), but we can expand it to reinforce the connection between microbiology and chemical results. (2) Alternatively, or in addition, we can revise the "Conclusions" section (pages 16-17, lines 724-727) to highlight this limitation and suggest that future studies incorporate microbiological analyses, such as metagenomic sequencing, to characterize the specific microbial populations in each type of fermentation vessel.
We are open to adjusting the manuscript according to the reviewer's recommendations and believe that these changes will strengthen the work's scientific relevance, aligning it with the academic community's expectations. We thank you again for your valuable observation, which will certainly contribute to improving the study.

Reviewer 3 Report (New Reviewer)
Comments and Suggestions for Authors
Dear authors,
You should add information whether the data is given for fresh mass or dry mass of the sample. Also, make sure that the authors you cite give the data for the same dry/fresh mass as you or recalculate them.
Below are the specific comments:
The title – it’s a question, therefore it should sound something like this: ‘Can different fermentation boxes improve the nutritional composition and the antioxidant activity of fermented and dried floodplain cocoa beans in the Brazilian Amazon?’
- 45 It sounds like the Brazil is the only country selling cocoa. Please, rewrite.
In the introduction there is no information whatsoever about fermentation boxes. As the manuscript is precisely about fermentation in these (different) boxes, the introduction should explain, why the boxes are used, what are the typical boxes and why, possibly, they could have an impact on the composition of the cocoa.
- 106-108 There is not enough information for the reproducibility of the extract production. Amount of sample, amount of solvent, time of the extraction, time of sonication, the ultrasonic bath model/power? Please, expand. Also, why is preparation of extracts in the sections about ‘chemicals and reagents’?
- 136-146 This section should be written clearer. You mention six days, 48 hours and then six days, and then seven days. Please, describe the procedure in a more clear way, like: ‘cocoa beans in boxes X, Y, Z were fermented for such and such time, in such and such conditions…; cocoa beans in box T were fermented in …; after the fermentation, cocoa beans from Y, Z were dried such and such;’. Right now this fragment could be read in many different ways, it’s not clear enough. Also, if the fermentation time is different, do you really analyse the influence of boxes, or fermentation methods? I think that the clarification is needed. Also, describe the boxes: material, dimensions?
- 150 Model of the mill, please, IKA produces a lot of them and, well, they are not the same.
- 182 If you want to use the ‘CAT’ for the first time, explain that it is ‘catechin’. But is the abbreviation really necessary?
- 187 Again, parameters or at least model of the sonication bath.
- 188 Filtered through what?
- 197-201 How much sample, how much ABTS solution, what was the absorbance of ABTS solution, what was the waiting time before measurement?
- 203-210 And how much of the acetate buffer and TPTZ solution was used…?
- 213 How much HCl? If 35%, then state that. Also, I assume it isn’t 35%, because, well, the HCl in 100% volume does not exist, so 65% HNO3 and 35% HCl doesn’t really seem plausible.
- 219 The results were compared using ANOVA, not the means. ANOVA does not directly compare means.
- 243-244 Almonds?
Table 1 – please, describe the abbreviations of boxes under the table, not over. Also, mean values, not means values. Also, its ‘ash’ not ‘ashes’. Also, change commas to dots (moisture, SB, standard deviation).
- 255 There is no need to add ‘with a mean standard deviation’.
- 260-261 Lack of oxygen can influence anaerobic conditions, not sugars and pH.
- 263-267 So, the PHB is a new box, while the others were used previously? Doesn’t it totally mitigate the sense of the study? You don’t analyse the influence of the box type, but the influence of the microflora of the box. Also, if some of the boxes were previously used and some weren’t, it should be clearly stated in the Materials and Methods section.
- 283 ‘Muted’ is not the proper word for this.
- 285 Doesn’t it just prove that the boxes with the microbiota (i.e. previously used) are better than new, unused boxes?
- 304, L. 311 Almonds, again. Correct everywhere. The manuscript is full of them.
- 322 What is PH sample? Nothing is mentioned about it in the manuscript and there is no data about the ash level of PH sample.
- 330-333 Reference for acceptable range for cocoa.
- 336 How does it indicate it? All the samples were fermented, right? There is no data about non-fermented cocoa beans, so… How can you determine, that it indicates it?
- 345-352 This seems misleading. The ‘cocoa fermentation’ process in a technological sense begins with a hydrolysis of proteins. Strictly speaking, process of fermentation is anaerobic metabolism of sugars, right? So, differentiate them, because a lot of the readers of this journal have stric biochemical background and use ‘fermentation’ as the microbial process, not technological process of cocoa fermentation. Also, almonds, again.
- 359 What literature? References.
- 365-368 Again, references. Again, almonds.
- 372-373 Does it? There is no information about the starting fat content in the cotyledons and the fat content in the peel, so you cannot state, that your data indicated it. You can, possibly, state that it ‘typically indicates’ that fact and use a reference.
- 376-377 What literature, again?
- 387-391 Again, literature.
- 393 Total phenolic compounds, not polyphenols. It’s Folin-Ciocalteu assessment.
- 397 when you use abbreviation, explain it firstly in the brackets.
- 422 Quantifying is simple. Comparing is complex.
- 422-426 Use past tense about the other authors. Highlighted, explained.
- 427-430 Also, references to you hypotheses.
- 434 Theobromine content, not results.
- 456-458 Again, references.
- 470-472 And how do you know that the polyphenol content was reduced by ~70%? The information about polyphenol content of the non-fermented beans was not given. If this is a general phenomenon – reference. Also, polyphenols or phenolic compounds?
Table 3 Flavonoids, not polyphenols.
- 503-505 You have already stated that.
- 506-511 Reference. The same in L. 529-536.
Table 4 There is no unit in the Table.
- 553 Why are they noteworthy? You are then citing, than they can be twice as high, so…?
- 556 Calcium? You were writing about potassium and magnesium.
- 566 Significant? What do you mean by that? Remember that significance is typical statistical term.
- 569-571 But what about the copper in the samples?
There really isn’t any discussion about most of the elements. You have mentioned what they are for and what is their concentration, but not described influence of the boxes (or, the process, as it should be called) on their content.
- 602 89.10% + 10.80% is not 100%.
Figure 2 – why have you decided to not show quartiles on the figure?
Author Response
Reviewers’ Comments to Authors
Referee: #3
- Comment: “Dear authors,
You should add information whether the data is given for fresh mass or dry mass of the sample. Also, make sure that the authors you cite give the data for the same dry/fresh mass as you or recalculate them.
Below are the specific comments:”
Our Reply: Thanks for this point. The raw material for this research is the fermented and dried cocoa beans only.
Comment: “The title – it’s a question, therefore it should sound something like this: ‘Can different fermentation boxes improve the nutritional composition and the antioxidant activity of fermented and dried floodplain cocoa beans in the Brazilian Amazon?’
Our Reply: The tile was modified. Thank you for your suggestion.
- Comment: “45 It sounds like the Brazil is the only country selling cocoa. Please, rewrite.’
Our Reply: The sentence was improved. Page 1, Lines 44-45.
- Comment: “In the introduction there is no information whatsoever about fermentation boxes. As the manuscript is precisely about fermentation in these (different) boxes, the introduction should explain, why the boxes are used, what are the typical boxes and why, possibly, they could have an impact on the composition of the cocoa.’
Our Reply: The information was added. Please check on Page 2 (Lines 71-81).
- Comment: “106-108 There is not enough information for the reproducibility of the extract production. Amount of sample, amount of solvent, time of the extraction, time of sonication, the ultrasonic bath model/power? Please, expand. Also, why is preparation of extracts in the sections about ‘chemicals and reagents’?”
Our Reply: The paragraph was improved on Page 4, Lines 166-172.
- Comment: “136-146 This section should be written clearer. You mention six days, 48 hours and then six days, and then seven days. Please, describe the procedure in a more clear way, like: ‘cocoa beans in boxes X, Y, Z were fermented for such and such time, in such and such conditions…; cocoa beans in box T were fermented in …; after the fermentation, cocoa beans from Y, Z were dried such and such;’. Right now this fragment could be read in many different ways, it’s not clear enough. Also, if the fermentation time is different, do you really analyse the influence of boxes, or fermentation methods? I think that the clarification is needed. Also, describe the boxes: material, dimensions?”
Our Reply: All the section was rewritten (Pages 3-4, Lines 144-157). Thank you.
- Comment: “150 Model of the mill, please, IKA produces a lot of them and, well, they are not the same.”
Our Reply: The information was added: model A11 from Ika (Page 4, Line 161).
- Comment: “182 If you want to use the ‘CAT’ for the first time, explain that it is ‘catechin’. But is the abbreviation really necessary?”
Our Reply: Thank you for your question. We believe that is important for readers differentiate catechin (CAT) from epicatechin (ECE) in the results.
- Comment: “187 Again, parameters or at least model of the sonication bath.”
Our Reply: The information was added previously (Page 4, Lines 166-172).
- Comment: “188 Filtered through what?”
Our Reply: The information was added (Page 4, Lines 183-184).
- Comment: “197-201 How much sample, how much ABTS solution, what was the absorbance of ABTS solution, what was the waiting time before measurement?”
Our Reply: The paragraph was improved with this information. Please, check on Page 5 (Lines, 218-223). Thank you.
- Comment: “203-210 And how much of the acetate buffer and TPTZ solution was used…?”
Our Reply: The paragraph was improved with this information. Please, check on Page 5 (Lines, 228-240). Thank you.
- Comment: “213 How much HCl? If 35%, then state that. Also, I assume it isn’t 35%, because, well, the HCl in 100% volume does not exist, so 65% HNO3 and 35% HCl doesn’t really seem plausible.”
Our Reply: Also, this information was added in the 2.4.4.3. Frap section. Pleach check on page 5 (Lines 228-240).
- Comment: “219 The results were compared using ANOVA, not the means. ANOVA does not directly compare means.”
Our Reply: The sentence was rewritten (Page 6, Lines 248-250).
- Comment. “243-244 Almonds?”
Our reply: “Almonds” word was deleted replaced for “cocoa beans” in all manuscript. Thank you.
- Comment. “Table 1 – please, describe the abbreviations of boxes under the table, not over. Also, mean values, not means values. Also, its ‘ash’ not ‘ashes’. Also, change commas to dots (moisture, SB, standard deviation).”
Our reply: The corrections were been made (Page 6).
- Comment. “255 There is no need to add ‘with a mean standard deviation’.”
Our reply: The sentence was improved (Page 7, Line 285).
- Comment. “260-261 Lack of oxygen can influence anaerobic conditions, not sugars and pH.”
Our reply: The paragraph was rewritten. Please, check on Page 7, Lines 289-298. Thank you.
- Comment. “263-267 So, the PHB is a new box, while the others were used previously? Doesn’t it totally mitigate the sense of the study? You don’t analyse the influence of the box type, but the influence of the microflora of the box. Also, if some of the boxes were previously used and some weren’t, it should be clearly stated in the Materials and Methods section.”
Our reply: The sentence was rewritten (Page 7, Lines 292-298).
- Comment. “283 ‘Muted’ is not the proper word for this.”
Our reply: We replaced this word for “soft” (Page 7, Line 314). Thank you.
- Comment. “285 Doesn’t it just prove that the boxes with the microbiota (i.e. previously used) are better than new, unused boxes?”
Our reply: As explained in previous answers, the hexagonal shape promoted an accumulation of cocoa beans in the corners of the box, which may have favored the lack of homogenization of the fermentation mass, thus providing an increase in the acidity levels of the fermented cocoa beans.
- Comment. “304, L. 311 Almonds, again. Correct everywhere. The manuscript is full of them.”
Our reply: “Almonds” word was deleted replaced for “cocoa beans” in all manuscript. Thank you.
- Comment. “322 What is PH sample? Nothing is mentioned about it in the manuscript and there is no data about the ash level of PH sample.”
Our reply: There was a typewritten mistake. The correct form is PHB (Page 8, Line 345).
- Comment. “330-333 Reference for acceptable range for cocoa.”
Our reply: The paragraph was improved. Please, check on Page 8, Lines 353-357. Thank you.
- Comment. “336 How does it indicate it? All the samples were fermented, right? There is no data about non-fermented cocoa beans, so… How can you determine, that it indicates it?”
Our reply: The paragraph was removed due to its predominantly empirical nature and the lack of direct factual evidence within this study. Although references could be included to provide justification, they would not sufficiently substantiate the claims. Thus, its exclusion does not compromise the overall quality of this paper. Thank you.
- Comment. “345-352 This seems misleading. The ‘cocoa fermentation’ process in a technological sense begins with a hydrolysis of proteins. Strictly speaking, process of fermentation is anaerobic metabolism of sugars, right? So, differentiate them, because a lot of the readers of this journal have stric biochemical background and use ‘fermentation’ as the microbial process, not technological process of cocoa fermentation. Also, almonds, again.”
Our reply: Thank you very much for your comment. We rewritten the paragraph for a better comprehension (Pages 8-9, Lines 372-388).
- Comment. “359 What literature? References.”
Our reply: We added Araujo et al. and Chagas Junior et al. (Page 9, Line 410).
Ref.:
Araujo, Q.R.; Fernandes, C.A.F.; Ribeiro, D.O.; Efraim, P.; Steinmacher, D.; Lieberei, R.; Bastide, P.; Araujo, T.G. Cocoa Quality Index—A proposal. Food Control 2014, 46, 49–54.
Chagas Junior, G.; Espírito-Santo, J.C.A.; Ferreira, N.R.; Marques-da-Silva, S.H.; Oliveira, G.; Vasconcelos, S.; Almeida, S.F.O.; Silva, L.R.C.; Figueredo, H.M.; Lopes, A.S. Yeast isolation and identification during on-farm cocoa natural fermentation in a highly producer region in northern Brazil. Sci. Plena 2020, 16, 121502.
- Comment. “365-368 Again, references. Again, almonds.”
Our reply: We changed “almonds” for “fermented cocoa beans” and added a reference (Page 9, Lines 413-414).
- Comment. “372-373 Does it? There is no information about the starting fat content in the cotyledons and the fat content in the peel, so you cannot state, that your data indicated it. You can, possibly, state that it ‘typically indicates’ that fact and use a reference.”
Our reply: The paragraph was improved (Page 9, Lines 408-412).
- Comment. “376-377 What literature, again?”
Our reply: This information is according to the Cocoa Index, published by Araujo et al.
Ref.:
Araujo, Q.R.; Fernandes, C.A.F.; Ribeiro, D.O.; Efraim, P.; Steinmacher, D.; Lieberei, R.; Bastide, P.; Araujo, T.G. Cocoa Quality Index—A proposal. Food Control 2014, 46, 49–54.
- Comment. “387-391 Again, literature.”
Our reply: The reference Chagas Junior et al. was added (Page 9, Line 428).
Ref.:
Chagas Junior, G.C.A.; Ferreira, N.R.; Lopes, A.S. The microbiota diversity identified during the cocoa fermentation and the benefits of the starter cultures use: An overview. Int. J. Food Sci. Technol. 2020, 1–9.
- Comment. “393 Total phenolic compounds, not polyphenols. It’s Folin-Ciocalteu assessment.”
Our reply: We improved the nomination for Total Phenolic Compounds (Page 9, Line 430).
- Comment. “397 when you use abbreviation, explain it firstly in the brackets.”
Our reply: We identified a mistake in the Total Polyphenols results. We recalculate and modified the information on Table 2 and in the manuscript (Page 10, Lines 435-438). Thank you.
- Comment. “422 Quantifying is simple. Comparing is complex.”
Our reply: The sentence was improved (Page 10, Lines 459-461).
- Comment. “422-426 Use past tense about the other authors. Highlighted, explained.”
Our reply: The sentence was improved (Page 10, Lines 459 and 461).
- Comment. “427-430 Also, references to you hypotheses.”
Our reply: The sentence was rewritten (Page 10, Lines 465-466).
- Comment. “434 Theobromine content, not results.”
Our reply: The denomination was changed for “content” (Page 10, Line 472). Thank you.
- Comment. “456-458 Again, references.”
Our reply: We added Araujo et al., in Line 496.
- Comment. “470-472 And how do you know that the polyphenol content was reduced by ~70%? The information about polyphenol content of the non-fermented beans was not given. If this is a general phenomenon – reference. Also, polyphenols or phenolic compounds?”
Our reply: The paragraph was removed due to its predominantly empirical nature and the lack of direct factual evidence within this study. Although references could be included to provide justification, they would not sufficiently substantiate the claims. Thus, its exclusion does not compromise the overall quality of this paper. Thank you.
- Comment. “Table 3 Flavonoids, not polyphenols.”
Our reply: Thank you. The word was modified.
- Comment. “503-505 You have already stated that.”
Our reply: Thank you for your suggestion. The paragraph was deleted.
- Comment. “506-511 Reference. The same in L. 529-536.”
Our reply: We added the references: Pinheiro et al. (Page 12, Line 543) and Gaspar et al. (Page 12, Line 568).
Ref.:
Pineiro, J.M.L.; e Silva, A.P.S.; Pantoja, K.R.P.; Cardoso, M.A.R.; de Azevedo, F.F.M.; de Melo, L.V.G.; Campos, G.I.A.; Mota, R.V.; Mouzinho, R.S.; Carvalho Junior, R.N. Supercritical extraction of butter from agroindustrial cocoa residue from the Amazon. J. Supercrit. Fluids 2025, 222, 106560, doi:10.1016/j.supflu.2025.106560.
Gaspar, D.P.; Chagas Junior, G.C.A.; de Aguiar Andrade, E.H.; Nascimento, L.D. do; Chisté, R.C.; Ferreira, N.R.; Martins, L.H. da S.; Lopes, A.S. How Climatic Seasons of the Amazon Biome Affect the Aromatic and Bioactive Profiles of Fermented and Dried Cocoa Beans? Molecules 2021, 26, 3759, doi:10.3390/molecules26133759.
- Comment. “Table 4 There is no unit in the Table.”
Our reply: We added the unit μg/g on the Table 4. Thank you.
- Comment. “553 Why are they noteworthy? You are then citing, than they can be twice as high, so…?”
Our reply: We rewritten the sentence (Page 13, Lines 585-586).
- Comment. “556 Calcium? You were writing about potassium and magnesium.”
Our reply: We written the sentence (Page 13, Lines 590-592).
- Comment. “566 Significant? What do you mean by that? Remember that significance is typical statistical term.”
Our reply: Thank you. We changed for “detectable values” (Page 13, Line 599).
- Comment. “569-571 But what about the copper in the samples?”
Our reply: We added some information about the role of copper in the cocoa fermentation (Pages 13-14, Lines 605-612). Thank you.
- Comment.. “There really isn’t any discussion about most of the elements. You have mentioned what they are for and what is their concentration, but not described influence of the boxes (or, the process, as it should be called) on their content.”
Our reply: We added some important information for the readers about the role of these microelements during the cocoa fermentation, on the manuscript (Page 14, Lines 635-647). We hope it does not interfere with the quality of the manuscript.
- Comment. “602 89.10% + 10.80% is not 100%.”
Our reply: We recalculated and corrected for 99.9% (Page 15, Line 670).
- Comment. “Figure 2 – why have you decided to not show quartiles on the figure?”
Our reply: This is a good point. Principal Component Analysis (PCA) is a widely applied statistical technique for reducing the dimensionality of datasets by identifying principal components that capture the majority of data variability. However, PCA does not directly utilize quartiles, as these are descriptive statistical measures used to assess data dispersion on a percentage scale (e.g., first, second, and third quartiles).
Once PCA is applied, the resulting principal components are linear combinations of the original variables and define new dimensions in the transformed space. Consequently, quartile-based measures, which are specific to individual variables, are not directly applicable to the interpretation of principal components. While PCA is primarily employed to reduce data complexity while preserving variance, quartiles serve as complementary statistical tools, typically used during preprocessing or in descriptive analysis stages. Additionally, quartiles are more commonly employed in the context of inferential statistical techniques such as ANOVA, rather than in PCA-based multivariate analyses.
Ref.:
Bardinet, J., Pouchieu, C., Chuy, V., Merle, B., Pellay, H., Lefèvre-Arbogast, S., ... & Féart, C. (2025). Association between nutrient patterns and odds of depressive symptomatology: a population-based cohort of older adults followed during 15-y. European Journal of Nutrition, 64(2), 1-12.
Yang, T. C., & Shoff, C. (2025). Social Vulnerability and Opioid Use Disorder Rate Among Older Medicare Beneficiaries in US Counties: How Has This Relationship Evolved Since Baby Boomers Entered Older Adulthood?. Public Policy & Aging Report, 35(1), 10-17.
Zahedi, H., Jowshan, M. R., Rasekhi, H., Amini, M., Sadeghi, O., Mehdizadeh, M., ... & Hajifathali, A. (2025). The association between the dietary inflammatory index and multiple myeloma: a case–control study. Scientific Reports, 15(1), 3123.

Round 2
Reviewer 2 Report (New Reviewer)
Comments and Suggestions for Authors
I accept the authors' motivation and the solution they proposed, as I understand that, at this stage, the absence of microbiological analyses limits the possibility of including and discussing related results. However, I hope that future studies they will replicate this research with a stronger focus on the microbiological aspects of cocoa bean fermentation, rather than exclusively on the type of fermentation boxes.
That said, I believe the manuscript still requires some changes before it can be considered for publication. In particular, the description of the fermentation process should be made more precise and scientifically rigorous. The following points should be addressed:
- L 290: The two-phase model of cocoa fermentation was proposed by De Vuyst and Leroy (ref. 33), not by Adler, as incorrectly stated.
- L 291–292: The first phase involves yeasts and lactic acid bacteria, not acetic acid bacteria, as currently reported.
- L 381–388: The description of the aerobic phase is incomplete. Specifically, the pivotal role of acetic acid bacteria is not addressed. I recommend revising this section based on Chagas et al. (2021) and De Vuyst and Leroy (2020) (ref. 15 and 33), which provide a more accurate explanation of this phase.
Specific comments:
- L62–63: Please amend “some yeast species” to “some yeast and bacterial species”.
- Table 2 legend: Please check the accuracy of the asterisks, as they may be incorrect or misleading.
- Reference 15 (Lines 795–796): The year of publication is incorrect and should be updated to 2021.
Author Response
Reviewers’ Comments to Authors
Reviewer: #2
- Comment: “I accept the authors' motivation and the solution they proposed, as I understand that, at this stage, the absence of microbiological analyses limits the possibility of including and discussing related results. However, I hope that future studies they will replicate this research with a stronger focus on the microbiological aspects of cocoa bean fermentation, rather than exclusively on the type of fermentation boxes.
That said, I believe the manuscript still requires some changes before it can be considered for publication. In particular, the description of the fermentation process should be made more precise and scientifically rigorous. The following points should be addressed:”.
Our reply: Thank you for your time for evaluate our manuscript.
- Comment: “L 290: The two-phase model of cocoa fermentation was proposed by De Vuyst and Leroy (ref. 33), not by Adler, as incorrectly stated.”
Our reply: Thank you for this point. The citation was corrected (Page 7, Line 292).
- Comment: “L 291–292: The first phase involves yeasts and lactic acid bacteria, not acetic acid bacteria, as currently reported.”
Our reply: The information was improved (Page 7, Line 293). Thank you.
- Comment: “L 381–388: The description of the aerobic phase is incomplete. Specifically, the pivotal role of acetic acid bacteria is not addressed. I recommend revising this section based on Chagas et al. (2021) and De Vuyst and Leroy (2020) (ref. 15 and 33), which provide a more accurate explanation of this phase.”
Our reply: Thank you very much for this. The paragraph was rewritten based in Chagas Junior et al., (2021) and De Vuyst and Leroy (2020). Please, check on Page 8 (Lines 386-397).
- Comment: “Specific comments:
- L62–63: Please amend “some yeast species” to “some yeast and bacterial species”.
Our reply: The sentence was improved (Page 2, Line 63).
- Comment: “Table 2 legend: Please check the accuracy of the asterisks, as they may be incorrect or misleading.”
Our reply: Thank you for this point. The asterisks in the legend of the Table 2 was improved (Page 10, Lines 451-455).
- Comment: “Reference 15 (Lines 795–796): The year of publication is incorrect and should be updated to 2021.”
Our reply: The year of the publication was corrected (Page 18, Line 805). Thank you.

Reviewer 3 Report (New Reviewer)
Comments and Suggestions for Authors
Dear authors,
the manuscript is significantly better. However, some small detail:
L. 150 Please, describe the average dimensions and material of HP and LSP, because it still is not clear for a reader, what was the experiment precisely about. Photograph would be also great, but if that's not possible, then, so it goes.
All the best.
Author Response
Reviewers’ Comments to Authors
Reviewer: #3
- Comment: “Dear authors,
the manuscript is significantly better. However, some small detail:
- 150 Please, describe the average dimensions and material of HP and LSP, because it still is not clear for a reader, what was the experiment precisely about. Photograph would be also great, but if that's not possible, then, so it goes.
All the best.”.
Our reply: Thank you for your time for evaluate our manuscript. The specification of HP and LSP fermentation boxes were added and can find on Pages 3 and 4 (Lines 149-154).
This manuscript is a resubmission of an earlier submission. The following is a list of the peer review reports and author responses from that submission.
Round 1
Reviewer 1 Report
Comments and Suggestions for Authors
Manuscript is not able to be published at the present form. Editing is required.
In general:
-authors must find suitable keywords.
-english must be improved.
-please check the numbers of the references.
-please unify the way to cite the references. according to Journal Guide for Authors
Comments on the Quality of English LanguageEnglish language must be improved.
Reviewer 2 Report
Comments and Suggestions for Authors
This study investigated the influence of different fermentation beds on the nutrient and antioxidant composition of várzea cocoa dry almond. The research maybe help to improve quality of cocoa beans. However, the writing is so terrible, the throughout check and revisions should be carried out before reconsideration. Questions and suggestions are as following.
1 the title is not accurate to include the main topic of this research.
2 the abstract did not provide the most important conclusion, and data processing and statistical analysis could not be included in this part.
3 keywords, important keywords are missing.
4 the fermentation technology and the effects on nutrients and phenolic compositions should be reviewed in Introduction, and the scientific hypothesis and objectives of this research should be provided.
5 reagents for the experiments should be provided in Materials and Methods, and methods should be detailed for reading and repeation of the experiments.
6 how to express your results for total flavonoid content, DPPH, ABTS, FRAP assays?
7 did you determine total phenolic contents of the samples?
8 the data processing and statistical analysis should be included in Materials and Methods.
9 so many mistakes should be checked, for example, citation style.
10 the tables are not standard.
11 how to conduct PCA, did you have PCA plots?
12 how to conduct correlation analysis? No significance was labeled in Table 7.
13 conclusion should be concise, and summarize your important research results, or put forward some shortcomings, not to discuss.
14 most of the references are outdated.
Reviewer 3 Report
Comments and Suggestions for Authors
The authors investigated the influence of different fermentation beds including projected hexagonal box - PHB, square box - SB, hamper - HP, and local square box - LSB on the nutrient and antioxidant composition of várzea cocoa (Theobroma cacao L.) dry almond from the fermentation to the drying step. The topic is very interesting and provides much information, and could atract the interest of readers of Foods. However, some issues should be improved.
The authors provided huge information about the nutrient and antioxidant composition of várzea cocoa (Theobroma cacao L.) obtained from different fermentation methods. However, I think the flavor of cocoa bean is very important, especially, after roasting, the authors did not investigate it, why?
I suggest the authors add the results of unfermented cocoa bean as the control.
Line 91, figure 10, please check.
Please change “minutes” to “min”.
Please check the symbol of Celsius degree.
Line 104, “A fourth method, called local square box (LSB), is used by farmers in the region”, I suggest the authors change “A” to “The”.
Line 124, Total Mineral Residue, suggested change: Total mineral residue
Please provide more information about sample and fermentation method, such as sample collection time, sample weight, diagram of fermentation bed, and how to ferment the almond.
Line 215, please check Fernandes et al. [2222].
Please check the format of all the tables, a three-line table is available.
Line 412, please check [20Error! Reference source not found.]
The conclusion is too long, please rewrite.
Comments on the Quality of English LanguagePlease improve English